# Bioavailability of Rosehip (*Rosa canina* L.) Infusion Phenolics Prepared by Thermal, Pulsed Electric Field and High Pressure Processing

**DOI:** 10.3390/foods11131955

**Published:** 2022-07-01

**Authors:** Gulay Ozkan, Tuba Esatbeyoglu, Esra Capanoglu

**Affiliations:** 1Department of Food Engineering, Faculty of Chemical and Metallurgical Engineering, Istanbul Technical University, Maslak, 34469 Istanbul, Turkey; ozkangula@itu.edu.tr; 2Department of Food Development and Food Quality, Institute of Food Science and Human Nutrition, Gottfried Wilhelm Leibniz University Hannover, Am Kleinen Felde 30, 30167 Hannover, Germany

**Keywords:** bioactive compounds, LC-MS, transepithelial transport, metabolic fate, Caco-2 cell culture, non-thermal processing

## Abstract

In this study, the *in vitro* bioavailability of rosehip infusion phenolics, mainly catechin, as a response to conventional and non-thermal treatments by combining gastrointestinal digestion and a Caco-2 cell culture model, was investigated. After application of thermal treatment (TT, 85 °C/10 min), high pressure (HPP, 600 MPa/5 min) or pulsed electric field (PEF, 15 kJ/kg) processing, all samples were subjected to simulated gastrointestinal digestion. Then, the amount of maximum non-toxic digest ratio was determined by the cytotoxicity sulforhodamine B (SRB) assay. Next, Caco-2 cells were exposed to 1:5 (*v*/*v*) times diluted digests in order to simulate the transepithelial transportation of catechin. Results showed that non-thermally processed samples (5.19 and 4.62% for HPP and PEF, respectively) exhibited greater transportation across the epithelial cell layer compared to than that of the TT-treated sample (3.42%). The present study highlighted that HPP and PEF, as non-thermal treatments at optimized conditions for infusions or beverages, can be utilized in order to enhance the nutritional quality of the final products.

## 1. Introduction

The fruit of rosehips (*Rosa canina*), having a reddish color, is mainly used in the dried form due to its limited period of availability, low stability during storage and the slightly sour taste of the fresh fruit [1]. Moreover, a variety of products including tea, juice, nectar, jam, marmalade, pestil [2] and vinegar [3], can be produced using rosehip. The great nutraceutical features being recognized in folk medicine arise from its high bioactive content, including vitamin C, carotenoids, tocopherols, phenolic acids, flavonoids, proanthocyanidins and tannins [4], as well as the abundance of minerals (calcium, phosphorus, potassium), pectin and essential oils [5].

Rosehip has been widely consumed to treat many diseases due to its numerous health benefits, which include antioxidant, antimicrobial, anti-inflammatory, anti-arthritic, analgesic, anti-diabetic, immunosuppressive, skin-ameliorative, cardio- and gastro-protective effects, as well as being a remedy for the common cold [1,6]. 

On the other hand, it should be considered that the health-promoting effects of bioactive compounds are mainly attributed to their bioavailability. The bioavailability and transportation of a digested compound throughout the intestinal barrier, and its transfer to blood circulation and tissues to exhibit its biological action, mainly depend on human transporters and phase I and phase II metabolism, the food matrix and food processing, among others [7,8,9]. In addition to these, the tendency of consumers to prefer minimally processed nutritious food products has brought about the exploration of alternative food processing techniques instead of conventional thermal treatments [10]. Thereby, due to the low bioavailability of phenolic compounds, it is necessary to carry out investigations on technological, biotechnological or nutritional approaches that could improve the transformation of phenolic substances into metabolites with higher bioavailability values. Technological treatments may trigger chemical or physical modifications in foods, with a positive impact on the bioaccessibility and thus bioavailability of phenolics. These alterations include the release of phenolic substances from the matrix due to changes in food structure, as well as the protection of phenolic compounds throughout the gastrointestinal tract due to interactions between different food components [7,11].

HPP and PEF applications, or their combinations with others such as mild heat treatment, ultrasound, ultraviolet light, high-intensity light pulses, manothermosonication, etc., have been identified as useful implementations in terms of not only improving the shelf-life (microbial quality) but also preserving the nutritive value and functional quality of foods [10,12,13]. In a study, the bioaccessibility of total flavonoids and chlorogenic acid content were found to be higher in HPP- and PEF-treated cranberrybush puree samples at specified operating conditions, in comparison to non-treated control samples. Furthermore, the recovery of chlorogenic acid in apical and basolateral compartments, as well as its bioavailability during transportation, were not negatively affected by non-thermal applications, and comparable results were obtained for both control and processed samples [7]. On the contrary, in some studies, traditional thermal treatments have been reported to reduce the bioaccessibility of bioactives. For example, Rodríguez-Roque et al. [11] reported the bioaccessibility of vitamin C, total phenolic acids and total flavonoids in fruit juice blends as 11.1, 17.2 and 15.5%, respectively, which were statistically lower than those of HPP- and high-intensity-PEF-treated samples. 

Although there have been various studies investigating the bioaccessibility of phenolics present in processed food products [7,14,15,16,17,18,19], there is a lack of information regarding the effects of thermal or non-thermal food processing on the bioavailability of phenolic compounds. Regarding the aforementioned explanations, the goals of this research were to determine the process’ effect on the stability, metabolic fate and transport efficiency of the rosehip infusion phenolics using a combined *in vitro* gastrointestinal digestion and Caco-2 cell model. Due to the loss of other phenolics in rosehip infusions throughout gastrointestinal digestion [15], and being one of the most abundant phenolic compounds in this fruit [3,15,20], catechin was monitored during transepithelial digestion in this research. Changes in the catechin content of rosehip infusions after processing and during gastrointestinal digestion have been reported previously by our research group [15]. While the catechin concentration in infusions obtained by TT, HPP and PEF treatments changed from 2487 to 3796 mg/100 g before digestion, most of this phenolic compound was lost after gastrointestinal digestion in the range of 51.8–61.0%, depending on the treatment.

In the study presented here, we examined (i) the phenolic profile of the rosehip infusion by UPLC–qTOF–MS/MS, (ii) the influence of processing on the stability and transportation of a rosehip phenolic—catechin—from the apical to basolateral compartments and (iii) the bioavailability of catechin from rosehip infusions as affected by processing throughout transepithelial transport.

## 2. Materials and Methods

### 2.1. Materials

Human colon adenocarcinoma cells (Caco-2) were provided by the German Collection of Microorganisms and Cell Cultures (Braunschweig, Germany). Growing medium reagents were supplied by Pan Biotech (Aidenbach, Germany). Trypan blue was purchased from Carl Roth (Karlsruhe, Germany). All reagents in this study were purchased from Sigma-Aldrich (Steinheim, Germany). Ultrapure water (Purelab flex 3; Veolia Water Technologies, Celle, Germany) was used for the analyses. All other reagents were of analytical or HPLC grade.

### 2.2. Production of Rosehip Infusions

Dried rosehips were provided by a local supplier in Bursa, Turkey. Before preparation of rosehip infusions, dried rosehip samples were ground using a blender (Blendtec Classic 575, Bad Homburg, Germany) at 22,000× *g* for 30 s. Ground samples were kept at ambient conditions in the dark. Rosehip infusions were prepared according to Ilyasoğlu and Arpa [21]. The weight of the rosehip in the infusions was fixed to 5% (*w*/*v*). After processing by thermal or non-thermal applications, the infusions were filtered by filter paper in order to remove the solid part. The filtrates were left at −80 °C (TSX50086V Ultra-Low Freezers, Thermo Fisher Scientific, Darmstadt, Germany) until *in vitro* bioaccessibility and cell culture analysis. 

### 2.3. Processing Techniques

TT, HPP and PEF processing parameters were selected based on a previous study [15], in which the maximum retention of total phenolics, total flavonoids, catechin content and antioxidant potential were obtained after HPP at 600 MPa for 5 min and PEF at 15 kJ/kg specific energy input generated by a pulse number of 135.

For TT, five grams of sample was mixed with 100 mL of water at 85 °C. Then, the beaker was surrounded by aluminum foil and held for 10 min without heating. After 10 min, the infusion was cooled and separated from the residue by using Whatman No. 4 filter paper. Infusions (filtrate) in the sterilized falcons were stored at −80 °C until *in vitro* bioaccessibility and cell culture analyses. 

The rosehip infusions were pressurized using industrial-scale equipment (Wave 6000/55, Hiperbaric S.A., Burgos, Spain). As a transmitting medium, water was used. Before treatment, the ground rosehip sample/water blend (5% (*w*/*v*)) was transferred into low-density polyethylene packages and vacuum-sealed. Then, these infusions were subjected to pressurization at 600 MPa for 5 min. At the end of the treatment, the infusion was separated from the residue by using Whatman No. 4 filter paper. The infusions (filtrates) in the sterilized falcons were kept at −80 °C until *in vitro* bioaccessibility and cell culture analyses.

In this study, a batch pilot-scale device (PEF Pilot, Elea GmbH, Quakenbrück, Germany) with a voltage of up to 30 kV was used. PEF treatment conditions were as follows: specific energy input of 15 kJ/kg at 1 kV electric field strength and pulse number of 135. After processing, the infusion was separated from the residue by using Whatman No. 4 filter paper. The infusions (filtrates) in the sterilized falcons were stored at −80 °C until *in vitro* bioaccessibility and cell culture analyses.

### 2.4. Simulation of Gastrointestinal Digestion

In order to reveal the metabolic fate of the polyphenols in the rosehip infusions, which were obtained by TT, HPP and PEF treatments, they were subjected to simulated gastrointestinal digestion. 

The *in vitro* gastrointestinal digestion was simulated regarding the procedures of Ozkan et al. [7] and Sessa et al. [22], with minor modifications. This assay involves two sequential stages: gastric digestion and bile salt/pancreatin digestion. For the gastric digestion, the samples containing 1.3 mg/mL final concentration of pepsin at pH 2 were placed in a water bath (GFL 1092, Burgwedel, Germany) at 37 °C and 100 rpm for a 2 h incubation period. To simulate intestinal digestion, gastric chyme was mixed with 0.175 mg/mL pancreatin and 1.10 mg/mL bile salts at pH 6.5, and then placed in a water bath at 37 °C and 100 rpm for 2 h incubation. At the end of the experiment, digested samples were cooled immediately by using an ice bath and then centrifugation was applied at 10,000 rpm, 4 °C, for 30 min (Megafuge 8R; Thermo Scientific, Darmstadt, Germany) to remove the residual fraction from the bioaccessible fraction.

Bioaccessible fractions of the digests were stored at −80 °C until cell culture study.

### 2.5. Caco-2 Cell Culture Study

Cells were grown in DMEM with 4.5 g/L glucose and stable glutamine, 20% FBS, 1% MEM NEAA, 100 U/mL penicillin and 100 µg/mL streptomycin. Then, they were kept at 37 °C in a humidified atmosphere with air and 5% CO_2_. A Volt-Ohm Meter (Millicell^©^ ERS-2 Millipore, Bedford, MA, USA) was used to monitor the cell monolayer integrity in terms of transepithelial electrical resistance (TEER) value. Caco-2 cells with passage numbers of less than 30 were used in this investigation [7,15].

### 2.6. Cytotoxicity Test

The cytotoxicity test was carried out in order to calculate the maximum non-toxic digest concentration for transport experiments. The seeded cells in 96-well plates were left to differentiate for 14 days after confluency. Then, these cells were exposed to rosehip digests at 1/5 or 1/10 (digest/HBSS, *v*/*v*) dilution ratios. The cytotoxic potential of digests on Caco-2 cells was assessed by SRB assay [23] after 4 h of incubation. After incubation, differentiated Caco-2 cells were subjected to fixation by transferring them to 50 μL of 50% TCA (trichloroacetic acid in ultrapure water) and stored at 4 °C for 45 min. After rinsing these well plates with tap water, they were left to air-dry. Thereafter, the cells in the 96-well plates were stained with 70 µL SRB (0.4% in 1% glacial acetic acid) and held for 15 min. These well plates were rinsed with glacial acetic acid (1% in ultrapure water) and left to air-dry. Afterwards, the absorbance of re-suspended stain in 200 μL of 10 mM Tris buffer was measured at 490 nm by an Infinite M200 UV–visible spectrophotometer (Tecan, Crailsheim, German) by eliminating the background reading at 620 nm.

### 2.7. Transepithelial Transportation

For the transepithelial transportation, cells seeded in inserts with 0.4 µm pore diameter (Sarstedt, Nümbrecht, Germany) at 4 × 10^4^ cell/well density in 6-well transwell plates were supplied with 2 mL of growing medium in the upper compartment and 2.5 mL of growing medium in the bottom compartment. Cells were left for development and differentiation to confluent monolayers for 21 days post-seeding. Differentiated enterocytes exhibiting stable TEER values were utilized in the transport tests. 

The transport trial was carried out based on Wu et al. [24]. Growing medium in the apical and basolateral compartments of transwells was different to HBSS; samples were preincubated for 1 h and then discharged. Bioaccessible fractions of the digested samples were diluted at 1/5 (digest/HBSS, *v*/*v*) ratio for the transport experiments, due to the fact that the viability of cells after 4 h of exposure to the digests in this condition was found to be higher than 85–90%, based on the outcomes of the SRB assay. Cells were kept at 37 °C in air with 5% CO_2_ for a 4 h incubation period. Throughout incubation, 200 µL samples from the apical and basolateral compartments were collected in 2 h intervals. TEER values of the cells were determined before the experiment, after the last sampling and 24 h after the incubation period. Samples were kept at −80 °C until identification by UPLC–qTOF–MS/MS.

### 2.8. Identification of Phenolic Compounds in Rosehip Infusions by UPLC–qTOF–MS/MS

Phenolic compounds in rosehip infusions were identified according to a previous study, with minor modifications [25]. UPLC–qTOF–MS/MS determination was conducted with a Waters Acquity UPLC system (Waters Co., Milford, MA, USA) with a Waters Q-ToF Premier mass spectrometer coupled with an electrospray ionization (ESI) source. Chromatographic elution was performed with a Waters Acquity UPLC BEH Phenyl (2.1 × 100 mm, 1.7 µm) column. Mobile phase consisted of formic acid/MQ water (1/1000, *v*/*v*; eluent A) and formic acid/acetonitrile (1/1000, *v*/*v*; eluent B). The conditions of the linear gradient were selected as: 0 min, 10% B; 0–6.48 min, 65% B; 6.48–6.77 min, 100% B; 6.77–8.60 min, 100% B; 8.60–8.70 min, 10% B. The flow rate and the injection volume were 0.4 mL/min and 10 μL, respectively. The column temperature was set as 35 °C, while the temperature of the autosampler was kept at 10 °C. ESI-MS analysis was conducted in negative mode. Collision energies of 5 and 40 V for low and high energy levels, respectively, were used for full-scan LC-MS in the *m*/*z* range 100−1500. Masslynx V4.1 software from Waters was used for the acquisition and integration of chromatograms.

### 2.9. Quantification of Rosehip Infusion Phenolics by UPLC-DAD

The quantification of phenolic compounds in rosehip infusions was conducted by UPLC based on Ozkan et al. [26], with minor changes. Digests obtained from apical and basolateral sides were analyzed by a Waters Acquity UPLC system (Waters Co., Milford, MA, USA) combined with a diode array detector (DAD). Chromatographic elution was performed with a Waters Acquity UPLC BEH Phenyl (2.1 × 100 mm, 1.7 µm) column. Mobile phase consisted of formic acid/MQ water (1/1000, *v*/*v*; eluent A) and formic acid/methanol (1/1000, *v*/*v*; eluent B). The conditions of the linear gradient were selected as: 0 min, 10% B; 0–6.48 min, 65% B; 6.48–6.77 min, 100% B; 6.77–8.60 min, 100% B; 8.60–8.70 min, 10% B. The flow rate and the injection volume were 0.4 mL/min and 10 μL, respectively. Catechin concentration was determined by using its authentic standard. The calibration curve was plotted in the range of 12–125 µg/mL. The results were expressed as mg catechin per 100 g dw (dry weight) of rosehip. Each calculation was performed in three replicates.

### 2.10. Statistical Analysis

Experiments were performed at least in triplicate. Results were stated as mean ± standard deviation. Statistical analysis was conducted with SPSS software (version 20.0, SPSS Inc. Chicago, IL, USA). Applications were evaluated and compared to each other by using one-way analysis of variance (ANOVA) followed by a Tukey post hoc test (*p* < 0.05).

## 3. Results

### 3.1. Identification of Polyphenols in Rosehip Infusions 

Detection of polyphenols in rosehip infusions was carried out by using the UPLC–qTOF–MS/MS identification method, by comparing the mass information previously indicated in the literature. Table 1 represents the detected phenolic compounds together with their accurate mass, molecular formula and the MS/MS fragment ions. The phytochemical profile of the samples involves flavan-3-ols (catechin and epicatechin), proanthocyanidins (procyanidin dimer and procyanidin trimer), flavonols (quercetin, quercetin-pentoside, quercetin-3-*O*-glucoside and rutin) and phenolic acids (chlorogenic acid, gallic acid, rosmarinic acid and coumaric acid).

### 3.2. Caco-2 Cytotoxicity 

Prior to the transport experiments, the SRB assay, in which the cellular viability is based on the cellular protein content, was performed. The protein contents of the Caco-2 cells treated with processed rosehip infusion digests are shown in Figure 1.

According to the results, digests with 1/5 and 1/10 dilution ratios did not exhibit a major cytotoxic effect. In general, the cell viability being greater than 85% indicated appropriate cell monolayer integrity without any cell loss from the monolayer. Thus, based on the outcomes of the cytotoxicity assay, it was decided to continue with the 1/5 dilution ratio. 

### 3.3. Transport Experiments

#### 3.3.1. Control of the Cell Integrity

The quality of the Caco-2 cell monolayer as monitored before the transportation as well as after 4 h and 24 h of the treatment for all digests is depicted in Table 2. Before incubation, initial TEER values of all conditions were found to be around 350 Ωcm^2^ and there was no change of more than 15% from the starting values after 4 h and 24 h.

#### 3.3.2. Catechin Transport

Transport experiments were conducted by loading the digests to the apical level of the cells. Then, digested samples were collected from the apical and basolateral side at different time points (2 and 4 h) of transportation. In order to analyze the metabolic fate of the polyphenols in rosehip infusions affected by food processing, the contents of polyphenols in the digests obtained from apical and basolateral compartments during transepithelial transportation were determined (Table 3). Since the other phenolics were not detected in the digests [15], calculations of the apical and basal recoveries as well as transport efficiency were performed based on catechin.

According to the results, the stability of catechin, which was reported as apical side recovery, was found to be in the range of 86 to 93% after 2 h of exposure to the digests, whereas it was recorded at around 68–69% after 4 h of incubation. Moreover, there was no statistically significant difference among samples. On the other hand, results of the transportation of catechin from apical to basolateral compartments, shown as basolateral side recovery, indicated the higher bioavailability of catechin in HPP- and PEF-treated samples in comparison to the TT-treated one. Similarly, the transport efficiency of catechin was found to be statistically higher in HPP- and PEF-treated samples compared to the TT-treated sample (*p* < 0.05).

## 4. Discussion

The polyphenol profile of rosehip infusions was found to be in line with those indicated previously, indicating that it contains flavan-3-ols, proanthocyanidins, flavonols and phenolic acids [4,29]. In detail, catechin is reported as the major phenolic compound of *R. canina* and its content varies as 347 mg/100 g in fresh fruit [20], 15.2 mg/L in rosehip juice and 5.7 mg/L in rosehip vinegar [3] and 2487–3796 mg/100 g rosehip infusions [15], depending on the type of the product to be obtained from rosehip. Moreover, rutin [30], ellagic acid, gallic acid [31], quercetin, chlorogenic acid and cinnamic acid [29] were also demonstrated as the most prominent phenolic compounds.

*In vitro* methods for the determination of the bioavailability of bioactive compounds have been widely applied as alternatives to *in vivo* procedures, due to the positive correlation between *in vitro* transepithelial transportation and *in vivo* absorption in humans [32] and animals [33]. The Caco-2 human colon carcinoma cell line has been widely used in order to estimate the intestinal uptake and metabolic fate of phenolic substances [7,24,34,35,36].

During transport studies, the cytotoxicity and monolayer integrity of Caco-2 cells should be controlled in order to maintain the cell viability. As shown by SRB outputs, it was deduced that the digest dilution ratio was suitable to protect the durability of the cell monolayer. Similar results were also found by Wu et al. [24], indicating that there was no significant variation in the protein content (SRB) of Caco-2 cells supplied with aronia juice containing or free (blank) digests. The quality of the Caco-2 cell monolayer was monitored by TEER measurement before transportation, immediately after the transportation trial and after 24 h of treatment for all digests, which were higher than the previously suggested value of 200 Ωcm^2^ [37]. Results of this experiment were in line with those obtained for cranberrybush [7], aronia [24] and black carrot [38] treated cells, in which the TEER values during transport experiments were preserved to a large extent.

Regarding the results of transport experiments, it is possible to indicate that the concentration of the main phenolic compound, catechin, crossing through the cellular monolayer, gradually increased in the basolateral side during 4 h of incubation. In addition to this, the transportation of catechin was altered in a positive manner by non-thermal processing. Indeed, food processing has a potent impact on the retention of bioactive compounds owing to the induction of potential modifications in the physicochemical features of phenolic compounds [39].

The results of this study are in agreement with those obtained in previous studies. For instance, Gonçalves et al. [40] reported that there was an increase in the passage of sweet cherry phenolics throughout the cellular monolayer depending on the transport time, and the quantity of epicatechin that crossed the cell monolayer was obtained to be 0.37%. Moreover, Toydemir et al. [36] indicated that the basolateral side recovery of epicatechin in sour cherry fruit and nectar samples changed in the range of 1.1–3.8% depending on ingredients such as sucrose and citric acid, as well as the processing applied. It was presented in the literature that the low absorption of catechin from the apical to basolateral direction is mainly attributed to efflux transporters in the small intestine. Efflux transporters, including multi-drug resistance protein 2 and P-glycoprotein, have a crucial function in the permeation and distribution of polyphenols [41].

Findings in the current research were found to be in line with the results of a recent study in which the bioavailability of catechin after oral administration to rats was evaluated [42]. According to the results of the *ex vivo* permeation assay, there was low permeation across the intestinal tissue, and it reached 12.5 μg after one hour. Moreover, according to the *in vivo* assay, the concentration of catechin was found to be 120 ± 3.97 ng/mL as the maximum in the plasma of rats after one hour. Similar trends were also obtained by Peng et al. [43], who investigated the fate of tea polyphenols. Outcomes of this research suggested that there was approximately 0.5–13.8% recovery of the ingested tea polyphenols present in plasma samples from rats.

In addition to the above studies, Athmouni et al. [44] conducted research in order to improve the bioavailability and absorption of catechin by formulating a catechin–phospholipid complex. The plasma catechin concentration in rats was monitored for 24 h and the results showed that the phospholipid complex raised the catechin’s bioavailability. Similar to this examination, Zagury et al. [45] noticed that the bioavailability of green tea catechin was doubled in a β-lactoglobulin-based delivery system in comparison to the free form by measuring the plasma catechin values in rats. While the bioavailability of free catechin was determined as 241 nM·h, it reached 497 nM·h after 6 h. In another study, the metabolic fate of catechin in sour mangosteen fruits was tested in an *in vivo* animal model using mice. According to the results, the maximum plasma epicatechin concentration in the peel and the catechin level in the rind were obtained to be 62.03 and 1.10 μg/mL, respectively. Findings also showed that epicatechin and catechin exhibited higher bioavailability among cinnamic acid, gallic acid, chlorogenic acid, syringic acid and coumaric acid [46].

With regard to investigating the bioavailability of catechin in humans, Mukai et al. [47] carried out a study to analyze the *in vivo* absorption of catechins in the chocolate food matrix by evaluating the plasma concentrations of five healthy 22-year-old women. Based on the outcomes of this trial, it could be concluded that epicatechin gallate and epigallocatechin gallate amounts in plasma were reduced by using this matrix, whereas there was no alteration in epicatechin and epigallocatechin values. Moreover, epicatechin gallate and epigallocatechin gallate were detected in their aglycon form in the plasma; in contrast, epicatechin and epigallocatechin were present as conjugated metabolites. The pharmacokinetic properties of epicatechin gallate and epigallocatechin gallate were also examined in rats. This work found that the results of the animal study were in line with those of the human study [47]. 

## 5. Conclusions

The aim of this study was to highlight the transport dynamics of rosehip phenolics using a gastrointestinal digestion/Caco-2 cell culture-based assay. Moreover, the effects of thermal and non-thermal food processing methods on the stability and bioavailability of phenolic compounds were evaluated and compared to each other. Both HPP and PEF applications at selected process intensities enhanced the absorption of catechin. Based on the results obtained here, these techniques can be reported as promising treatments in order to produce infusions with higher nutritional value. On the other hand, it is a necessity to optimize the operating conditions of each treatment for each product. In addition to these, a variety of attempts, such as using delivery systems for bioactive compounds and food formulation studies, can be implemented in order to provide enhanced bioavailability of phenolic compounds. Moreover, the metabolic fate of the rosehip infusion phenolics may be explained by examining the sulfated, glucuronidated or methylated metabolites of phenolics after transportation through the gut epithelium, as well as after colonic fermentation.

## Figures and Tables

**Figure 1 foods-11-01955-f001:**
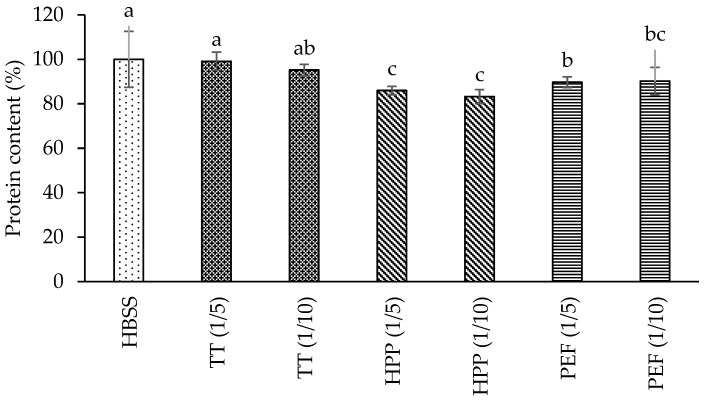
Cell viability expressed as percentage (%) compared to the sample-free (HBSS) cells for the 1/5 and 1/10 diluted rosehip samples determined by the SRB assay. The data presented in this figure consist of average values ± standard deviation of three independent batches. Different small letters on the columns indicate statistically significant differences between treatments (*p* < 0.05).

**Table 1 foods-11-01955-t001:** Identification of the major polyphenols detected in the rosehip infusions using UPLC–QTOF–MS/MS.

RT (min)	Compound	Molecular Formula	Theoretical Ion [M − H]^−^ *m*/*z*	RT (min)	Compound
1.17	Quercetin-pentoside	C_20_H_18_O_11_	433	301	[27]
1.39	Chlorogenic acid	C_16_H_18_O_9_	353	135, 161, 179, 191	[15]
1.93	Gallic acid	C_7_H_6_O_5_	169	125	[28]
2.56	Rosmarinic acid	C_18_H_15_O_8_	359	161, 179, 197	[28]
2.88	Catechin	C_15_H_14_O_6_	289	109, 125, 203, 245	[4]
2.96	Procyanidin dimer	C_30_H_26_O_12_	577	289, 407, 425, 451	[28]
3.20	Procyanidin dimer	C_30_H_26_O_12_	577	289, 407, 425, 451	[28]
3.31	Procyanidin trimer	C_45_H_38_O_18_	865	289, 577	[28]
3.57	Coumaric acid	C_9_H_8_O_3_	163	119	[29]
4.73	Quercetin-3-*O*-glucoside	C_21_H_20_O_12_	463	301	[4]
4.97	Rutin	C_27_H_30_O_16_	609	301	[29]
5.38	Quercetin	C_15_H_10_O_7_	301	151	[4]

**Table 2 foods-11-01955-t002:** TEER (Ωcm^2^) measurements of Caco-2 cells treated with rosehip infusion digests in HBSS (1/5).

Samples	After 4 h Incubation with Digests	After 24 h Incubation
HBSS (without digest)	300 ± 0.00 (86%)	355 ± 7.07 (101%)
TT	315 ± 21.2 (90%)	325 ± 35.4 (93%)
HPP	300 ± 0.00 (86%)	300 ± 0.00 (86%)
PEF	300 ± 0.00 (86%)	300 ± 0.00 (86%)

Numbers in brackets show the maintained TEER values as percentages.

**Table 3 foods-11-01955-t003:** Apical recovery, basal recovery and transport efficiency of catechin.

Sample	2 h			4 h		
	Apical Recovery (%) ^A^	Basal Recovery (%) ^B^	Transport Efficiency ^C^	Apical Recovery (%)	Basal Recovery (%)	Transport Efficiency
TT	86 ± 7 ^a^	2.65 ± 0.20 ^b^	0.031 ± 0.003 ^a^	68 ± 5 ^a^	3.42 ± 0.76 ^b^	0.050 ± 0.004 ^b^
HPP	90 ± 6 ^a^	3.10 ± 0.07 ^b^	0.035 ± 0.002 ^a^	68 ± 1 ^a^	5.19 ± 0.36 ^a^	0.076 ± 0.017 ^a^
PEF	93 ± 2 ^a^	3.29 ± 0.40 ^a^	0.035 ± 0.001 ^a^	69 ± 0 ^a^	4.62 ± 0.01 ^a^	0.067 ± 0.001 ^a,b^

Different small letters in the columns represent statistically significant differences (*p* < 0.05). ^A^ Apical side recovery percentages were calculated as (catechin concentration at the apical side after transport)/(catechin concentration at the apical side at 0 h of incubation) × 100. ^B^ Basolateral side recovery percentages were calculated as (catechin concentration at the basolateral side after transport)/(catechin concentration at the apical side at 0 h of incubation) × 100. ^C^ Transport efficiency of catechin was calculated as (basolateral side recovery, %)/(apical side recovery, %).

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
