# Peer review of "Bioavailability of Rosehip (Rosa canina L.) Infusion Phenolics Prepared by Thermal, Pulsed Electric Field and High Pressure Processing"

_foods, 2022, doi:10.3390/foods11131955_

Round 1

Reviewer 1 Report

I reviewed the manuscript entitled, Bioavailability of Rosehip (Rosa Canina L.) Infusion Phenolics Prepared by Thermal, Pulsed Electric Field and High Hydro-static Pressure Applications. Although the concept work seems novel, the introduction is poorly written. Methodology should be cited with appropriate literature. Discussion should be improved with reasoning.  

Abstract should be revised. Results section in the abstract seems incomplete. Authors should focus research highlights and recommend which treatment is good, with or without treatment?

Line 37: for first instance introduce what is HPP and PEF?

Introduction is poorly written

Authors should consider to revise completely by focusing on plant bioactive compounds, bioavailability, need of using alternative food processing technologies along with supported literature, research hypothesis and clear objective. As such, there is no clear information on why the authors have conducted this study.

Line 58: Turkiye Is not an English word. Please revise to English to attract international readers.

Like 65: in vitro or in-vivo should be in Italics throughout the manuscript

TT, HPP and PEF should be defined first instance

2.5. Caco-2 Cell Culture Study: please cite appropriate reference

2.8. Identification of Phenolic Compounds in Rosehip Infusions by UPLC-qTOF-MS/MS: please cite reference

Figure 1. indicate what are lower-case letters in footnote

What is the amount of catechin in infusion? Why did the authors only focus on catechin? The reason should be introduced in the introduction section.

References

All references should be revised according to journal format. Scientific names must be in Italics throughout the manuscript.

Author Response

Reviewer 1:

Comment: I reviewed the manuscript entitled, Bioavailability of Rosehip (Rosa Canina L.) Infusion Phenolics Prepared by Thermal, Pulsed Electric Field and High Hydro-static Pressure Applications. Although the concept work seems novel, the introduction is poorly written. Methodology should be cited with appropriate literature. Discussion should be improved with reasoning. 

Comment: Abstract should be revised. Results section in the abstract seems incomplete. Authors should focus research highlights and recommend which treatment is good, with or without treatment?

Response: We revised the results section in the abstract.

Comment: Line 37: for first instance introduce what is HPP and PEF?

Response: Explanations for HPP and PEF are written in abstract section for the first time.

Comment: Introduction is poorly written

Authors should consider to revise completely by focusing on plant bioactive compounds, bioavailability, need of using alternative food processing technologies along with supported literature, research hypothesis and clear objective. As such, there is no clear information on why the authors have conducted this study.

Response: We revised the introduction part by adding information about the bioavailability, need of using alternative food processing technologies along with supported literature, research hypothesis and clear objective.

Comment: Line 58: Turkiye Is not an English word. Please revise to English to attract international readers.

Response: We revised it.

Comment: Like 65: in vitro or in-vivo should be in Italics throughout the manuscript

Response: We corrected all of them.

Comment: TT, HPP and PEF should be defined first instance

Response: Explanations for TT, HPP and PEF are written in abstract section for the first time.

Comment: 2.5. Caco-2 Cell Culture Study: please cite appropriate reference

Response: We add an appropriate reference in this part.

Comment: 2.8. Identification of Phenolic Compounds in Rosehip Infusions by UPLC-qTOF-MS/MS: please cite reference

Response: There has already been a reference in this part (Kamiloglu, S.: Effect of different freezing methods on the bioaccessibility of strawberry polyphenols. Int. J. Food Sci. Technol. 2019, 54, 2652–2660. https://doi.org/10.1111/ijfs.14249).

Comment: Figure 1. indicate what are lower-case letters in footnote

Response: We added an explanation under Figure 1.

Comment: What is the amount of catechin in infusion? Why did the authors only focus on catechin? The reason should be introduced in the introduction section.

Response: We added detail information about this issue in the introduction part.

Comment: References

All references should be revised according to journal format. Scientific names must be in Italics throughout the manuscript.

Response: We checked all of them.

Reviewer 2 Report

Bioavailability of Rosehip (Rosa Canina L.) Infusion Phenolics Prepared by Thermal, Pulsed Electric Field and High Hydrostatic Pressure Applications

The introduction to the article should be a bit more elaborate. Similar studies are found in the literature, for example:

Impact of pilot-scale processing (thermal, PEF, HPP) on the stability and bioaccessibility of polyphenols and proteins in mixed protein- and polyphenol-rich juice systems. Uri Lesmesb, Andreas Juadjura, Volker Heinza, Cornelia Rauhc, Avi Shpigelmanb, Kemal Aganovica Innovative Food Science & Emerging Technologies. Volume 64, August 2020, 102426

The introduction should emphasize the innovativeness of the presented research.

The results could be presented in an interesting graphic form.

Author Response

Comment: The introduction to the article should be a bit more elaborate. Similar studies are found in the literature, for example:

Impact of pilot-scale processing (thermal, PEF, HPP) on the stability and bioaccessibility of polyphenols and proteins in mixed protein- and polyphenol-rich juice systems. Uri Lesmesb, Andreas Juadjura, Volker Heinza, Cornelia Rauhc, Avi Shpigelmanb, Kemal Aganovica Innovative Food Science & Emerging Technologies. Volume 64, August 2020, 102426

Response: We improved the introduction part. We also used this paper.

Comment: The introduction should emphasize the innovativeness of the presented research.

Response: We added detail information about this issue in the introduction part.

Comment: The results could be presented in an interesting graphic form.

Response: We tried to convert it into a graphic but since the data was not well represented, we preferred to keep it as a table.

Reviewer 3 Report

Bioavailability of Rosehip (Rosa Canina L.) Infusion Phenolics  Prepared by Thermal, Pulsed Electric Field and High Hydro static Pressure Applications

Comments:

Line no 22:  Abstract is not very well written and not explanatory.

Line no 28: Try to write the information in an linked manner, first give its properties like antioxidant and others then relate with health.

Line no 29: Which health property is more prominent?

Line no 33: Bioactive compounds availability mainly depends upon which factors?

Line no 37:  You should check these papers for updated data, these are comparative studies.

Food Chemistry, 2022, 371, 130821

International Journal of Food Science and Technology, 57(2), 816-826.

Innovative Food Science & Emerging Technologies, 2016, 38, 349-355

LWT-Food Science and Technology, 141(9), 111828.

Food Research International, 2021, 140, 110040.

Introduction is too short. Please add such type of studies which are on combined treatment of nonthermal techniques.

67. 2.3. Processing Techniques. I think processing conditions are not explained well. Please explain these treatments.

Line no 193: Graphical representation is appreciable.

Line no 198: Which component increase monolayer integrity?

Line no 236: Discussion does not include any data according to techniques.

Line no 286: Also discuss which technique impacts are more preferable? Results are not clear.

Main flaw:

Reasoning and comparison should be done and should be taken from those type of papers which are on comparison of thermal and non-thermal.  

Comparison and reasoning is the weaker section.

Author Response

Comment: Line no 22:  Abstract is not very well written and not explanatory.

Response: We improved the abstract part.

Comment: Line no 28: Try to write the information in an linked manner, first give its properties like antioxidant and others then relate with health.

Response: We revised the introduction part.

Comment: Line no 29: Which health property is more prominent?

Response: We revised this part as “Rosehip has been widely consumed against many diseases due to its numerous health benefits, which are antioxidant, antimicrobial, anti-inflammatory, anti-arthritic, analgesic, anti-diabetic, immunosuppressive, skin ameliorative, cardio- and gas-tro-protective effects as well as remedy for common cold.” We tried to emphasize all health effects that have been reported in the literature.

Comment: Line no 33: Bioactive compounds availability mainly depends upon which factors?

Response: We revised this part as “The bioavailability, transportation of digested compound throughout the intestinal barrier and transferring to blood circulation and tissues to exhibit its biological action, is mainly depend on human transporters and phase I and phase II metabolism, food matrix and food processing, among others.”

Comment: Line no 37:  You should check these papers for updated data, these are comparative studies.

Food Chemistry, 2022, 371, 130821

International Journal of Food Science and Technology, 57(2), 816-826.

Innovative Food Science & Emerging Technologies, 2016, 38, 349-355

LWT-Food Science and Technology, 141(9), 111828.

Food Research International, 2021, 140, 110040.

Response: Thank you very much. We improved the introduction part. We also used these papers.

Comment: Introduction is too short. Please add such type of studies which are on combined treatment of nonthermal techniques.

Response: We improved the introduction part.

We also add such information “HPP and PEF applications or their combinations with others such as mild heat treatment, ultrasound, ultraviolet light, high intensity light pulses, manothermosoni-cation, etc., have been identified as useful implementations in terms of not only increasing the shelf-life (microbial quality) but also preventing the nutritive value and functional quality of foods”.

Comment: 67. 2.3. Processing Techniques. I think processing conditions are not explained well. Please explain these treatments.

Response: We add such information “TT, HPP and PEF processing parameters were selected based on a previous study [6], in which the maximum retention of total phenolics, total flavonoids, catechin contents and antioxidant potentials were obtained after HPP at 600 MPa for 5 min and PEF at 15 kJ/kg specific energy input generated by 135 pulse number.”

Comment: Line no 193: Graphical representation is appreciable.

Response: We tried to convert some of the data in the table into a graphic but since the data was not well represented, we preferred to keep it as a table.

Comment: Line no 198: Which component increase monolayer integrity?

Response: We revised this sentence as “In general, the cell viability to be greater than 85% indicated an appropriate cell monolayer integrity without any cell loss from the monolayer”. Because, monolayer integrity of the cells forms during development and differentiation for 21 days post-seeding as indicated in section 2.7.

Comment: Line no 236: Discussion does not include any data according to techniques.

Response: We discussed the results in discussion part and we also improved this section.

Comment: Line no 286: Also discuss which technique impacts are more preferable? Results are not clear.

Response: “Both HPP and PEF applications at selected process intensities enhanced the absorption of catechin.”

Comment: Main flaw:

Reasoning and comparison should be done and should be taken from those type of papers which are on comparison of thermal and non-thermal.  

Comparison and reasoning is the weaker section.

Response: We improved the discussion part.

Round 2

Reviewer 1 Report

The quality of the manuscript is now improved. However, references are still not according to journal format. For example, journal name should be in Italics 

Author Response

Comment: The quality of the manuscript is now improved. However, references are still not according to journal format. For example, journal name should be in Italics.

Response: Thank you very much for your valuable suggestions. We revised the journal name to be in Italics and the year of the publication to be bold.

Reviewer 3 Report

Thanks a lot for your revisions 

Author Response

Comment: Thanks a lot for your revisions.

Response: Thank you very much for your valuable suggestions.